# ClimbQ: Class Imbalanced Quantization Enabling Robustness on Efficient Inferences

**Ting-An Chen**[1,2]**, De-Nian Yang**[2,3] **Ming-Syan Chen**[1,3]
[1]Graduate Institute of Electrical Engineering, National Taiwan University, Taiwan
[2]Institute of Information Science, Academia Sinica, Taiwan
[3]Research Center for Information Technology Innovation, Academia Sinica, Taiwan
tachen@arbor.ee.ntu.edu.tw, dnyang@iis.sinica.edu.tw, mschen@ntu.edu.tw

## Abstract

*Quantization* compresses models to low bits for efficient inferences which has received increasing attentions. However, existing approaches focused on balanced datasets, while imbalanced data is pervasive in the real world. Therefore, in this study, we investigate the realistic problem, quantization on class-imbalanced data. We observe from the analytical results that quantizing imbalanced data tends to obtain a large error due to the differences between separate class distributions, which leads to a significant accuracy loss. To address this issue, we propose a novel quantization framework, *Class Imbalanced Quantization (ClimbQ)* that focuses on diminishing the inter-class heterogeneity for quantization error reduction. ClimbQ first scales the variance of each class distribution and then projects data through the new distributions to the same space for quantization. To guarantee the homogeneity of class variances after the ClimbQ process, we examine the quantized features and derive that the homogeneity satisfies when data size for each class is restricted (bounded). Accordingly, we design a *Homogeneous Variance Loss (HomoVar Loss)* which reweights the data losses of each class based on the bounded data sizes to satisfy the homogeneity of class variances. Extensive experiments on class-imbalanced and benchmark balanced datasets reveal that ClimbQ outperforms the state-of-the-art quantization techniques, especially on highly imbalanced data.

## 1 Introduction

Convolutional neural networks (CNNs) with high computational complexity inhibits the deployment on resource-limited mobile devices [1, 2, 3, 4, 5]. Quantization has been studied an efficient method for reducing memory storage and accelerating inferences by compressing models to low bits [6, 7, 8, 9, 10, 11, 12]. The existing quantization approaches are generally developed under a common assumption that the data is balanced for separate classes. However, class-imbalanced data is ubiquitous in the real world.

Despite a realistic problem, quantization on class-imbalanced data has not been studied. Therefore, in this research, we investigate the issues of quantization on class-imbalanced data. The exploration results on CIFAR-10-LT (long-tailed CIFAR-10) are depicted in Fig. 1. Fig. 1a illustrates the data size of each class, where the largest class (indexed 0) has 50 times more data than the smallest class (indexed 9), i.e., an imbalanced dataset. We then conduct experiments to quantize the class-imbalanced data by exploiting a baseline quantization approach, uniform quantization. Fig. 1b shows that the majority classes (indexed 0 to 3) are quantized to 2 bits with only 2% accuracy loss (compared to the 32-bit model), whereas the minority classes (indexed 8 and 9) obtain 14% accuracy drop. It indicates that the quantization result is inclined to be dominated by the majority classes, which may result from the heterogeneity of class distributions as presented in Fig. 1c. Fig. 1c manifests that

36th Conference on Neural Information Processing Systems (NeurIPS 2022).

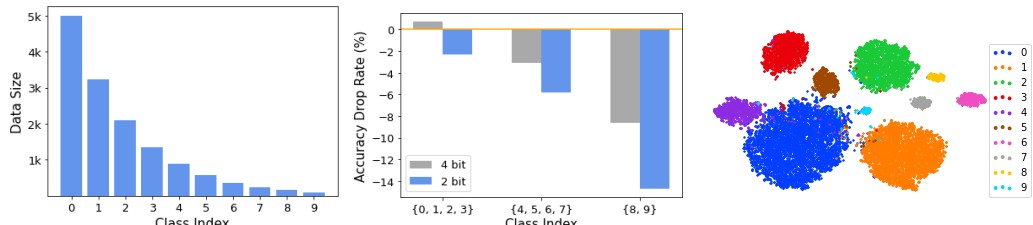

(a) Data Size of CIFAR-10-LT.  (b) Acc. Drop under Quantization.  (c) T-SNE of Features.

Figure 1: Experiment results on CIFAR-10-LT. Fig. 1a presents the data size of each class in CIFAR-10-LT (long-tailed CIFAR-10), where the maximum class has 50× data more than the minimum class. Fig. 1b depicts the rate of accuracy loss under quantization in comparison to the 32-bit model, where the majority classes have the least accuracy drop, while the accuracy of minority classes decreases the most (as much as 14% accuracy drop for a 2-bit model). It indicates that the majority classes determine the quantization result due to the fact illustrated in Fig. 1c that the majority classes not only have a large amount of data but also distributed in large variations.

the majority classes always have greater data variations than the minority classes. Therefore, the quantization results are inclined to be determined by the majority classes, which may lead to a large quantization error and a notable accuracy loss for the minority classes (validated in Fig. 1b).

Motivated by this challenge in quantization, we propose a *Class Imbalanced Quantization (ClimbQ)* scheme to reduce the differences between class distributions for a robust quantization on imbalanced data. According to the result as presented in Fig. 1c that the majority classes are distributed more diversely than the minority classes, *ClimbQ* scales the class distributions, allowing the class variations to be closer. Afterward, we project data through the new class distributions to the same uniform space for quantization to further decrease the quantization error[1].

To further ensure that the class distributions after quantization are homogeneous, we analyze the homogeneity of class variances based on *Levene's* hypothesis testing in the theory of statistics [13]. As our contribution, we demonstrate that the data size of each class is restricted to satisfy the homogeneity of class distributions (variances) and derive the bounds of the data sizes. Accordingly, we design a *Homogeneous Variance Loss (HomoVar Loss)* to rebalance the data of each class. We place emphases on the (minority) classes with a data size below the lower bound and weight down the loss of the (majority) classes with a larger data size over the upper bound to prevent the majority classes from dictating the quantization results.

Experiments demonstrate that ClimbQ+ (ClimbQ incorporated with HomoVar loss) outperforms the prior works on both imbalanced and balanced datasets. ClimbQ+ improves the state-of-the-art quantization approaches by 6% for 2-bit ResNet-56, while by 12% for 2-bit MobileNet-V2 on CIFAR-100-LT in highly imbalanced instances.

Our contributions are summarized as follows:

1. We develop a novel quantization scheme, *ClimbQ*, that makes the first attempt to reduce the differences between class distributions for a robust quantization process on imbalanced data.

2. We derive the bounds of data size for each class to guarantee the homogeneity of class variances. Accordingly, we propose a *HomoVar Loss* to reweight the loss on each class in order to prevent the majority classes from dictating the quantization results.

3. Experimental results demonstrate that ClimbQ+ (ClimbQ with HomoVar loss) has superior improvements over the state-of-the-art quantization on class-imbalanced and benchmark balanced datasets, especially significant for the low-bit models on highly imbalanced data.

## 2 Related works

**Quantization.** Quantization has received increasing attentions in recent years with the development of the Internet of Things (IoT), meeting the requirements of efficient computing and storage.

---

[1]We adopt the uniform quantization scheme due to its efficiency when deployed on hardware devices [6, 5].

(a) Class Imbalanced Quantization

(c) Learning Modules

$x_k \sim f_k$: $N(\mu_k, \sigma_k^2)$ → [Class Distribution Scaling & Generation] → $x'_k \sim f'_k$: $N(\mu_k, c_k^2 \sigma_k^2)$ → [Quantization, $Q$]

[Quantized CNN]

$Q(x'_k)$ — $w$

FC

$\hat{y}$ ⊗ $\mathcal{L}_H$

$y$ — $\mathcal{L}$

(b) Rebalancing for Homogeneity of Variances

$Q(x'_k)$; $n_k$ → [Appropriate Class Data Size Estimation] → $w(n_L ; n_U)$ → [HomoVar Loss, $\mathcal{L}_H$]

Figure 2: Overview of *ClimbQ*. Fig. 2a illustrates ClimbQ (CLass IMBalanced Quantization), which reduces differences between class distributions by scaling the standard deviation $\sigma_k$ by $c_k$. Afterward, we project the data to the uniform space for quantization $Q$ (detailed in Sec. 3.1). The ClimbQ algorithm is applied to each convolutional layer as shown in Fig. 2c. Furthermore, as presented in Fig. 2b, we derive the bounds $n_L$ and $n_U$ that satisfy the homogeneity of class variances by examining the quantized features under ClimbQ. Accordingly, we propose a *HomoVar* loss $\mathcal{L}_H$ to rebalance the classes by weighting. We mainly focus on the classes with data sizes out of the bounded range (introduced in Sec. 3.2). Fig. 2c shows that the HomoVar loss $\mathcal{L}_H$ is computed by multiplying the weight $w$ and data loss $\mathcal{L}$ for each class (illustrated as blue and green modules).

*Quantization-aware training (QAT)* is one branch of approaches. It mainly focuses on learning a clipping range [6, 14, 5, 15] or a data transformation [7, 8, 11, 16] for quantization with a small loss of accuracy. The well-trained statistics are then stored and utilized in low-bit inferences. In contrast, *Zero-shot quantization (ZSQ)* has recently been proposed to incorporate the concept of knowledge distillation, transferring the information from 32-bit models to the quantized networks [17, 10, 18, 19, 20]. They intend to reduce the feature differences between full-bit and low-bit models. Existing QAT or ZSQ methods learned a single quantization scheme for all data, implicitly assuming that the data follows identical distributions [5, 14, 9, 17, 18, 10, 11, 16]. However, class-imbalanced data is a realistic case in which the class distributions are diverse. It is visualized in Fig. 1c that the majority classes are always with greater variations and sizes, dominating the quantization results and resulting in a significant quantization error for the minority classes, as seen in Fig. 1b. Therefore, we aim to develop a robust quantization scheme on imbalanced data with a reduced quantization error. In particular, this work mainly follows the QAT research instead of ZSQ which exploits additional knowledge from a floating-point model and suffers from higher memory costs.

**Class-imbalanced learning.** Class-imbalanced problem poses a great challenge in deep learning, where the minority classes with few data generally have an inferior classification performance to the majority classes [21, 22, 23, 24]. Previous research has studied *resampling* to balance the classes to address this issue [25, 26, 27, 21]. They focus on collecting more data from the minority classes (i.e., over-sampling) or eliminating some data from the majority (i.e., under-sampling). Nevertheless, over-sampling increases memory storage and training time, whereas under-sampling raises the over-fitting problem since models are only learned from the sampled data [28]. Another branch of class-imbalanced learning approaches developed is *reweighting*. In contrast, they reweight the data losses of each class for balancing instead of sampling, which has no problems of memory consumption and over-fitting. Prior research suggested focusing more on the minority and assign small losses to the majority [29, 30, 31, 32, 33]. Nevertheless, they are only concerned with class data sizes. In this research, we design a reweighting loss that leverages not only the data size but also the quantized features to balance data for homogeneity of class variances with a reduced error during quantization.

# 3 ClimbQ

Fig. 2 introduces ClimbQ, a novel quantization scheme that considers the heterogeneity of class distributions on class-imbalanced data to reduce quantization errors. ClimbQ first scales class distributions that enables the new distributions with closer variances for quantization (depicted in Fig. 2a and detailed in Section 3.1). In Sec. 3.2, we investigate the homogeneity of class variances based on quantized features. From the analytical results, we derive the bounds of data sizes that satisfy the homogeneity condition (see blue modules in Fig. 2b and Fig. 2c). In Sec. 3.3, we design a *HomoVar* loss that reweights the data losses of each class according to the quantized data and the class data size (see green modules in Fig. 2c).

## 3.1 Class imbalanced quantization (ClimbQ)

Existing works quantized data from different classes under the same process, ignoring the heterogeneous class distributions in the real-world imbalanced data. As visualized in Fig. 1c on CIFAR-10-LT, the majority classes have greater variations than the minority classes, raising the concern that the quantization results incline to be bias toward the majority. As a result, the minority classes are quantized with inevitable biases that result in a significant accuracy loss (validated in Fig. 1b). To reduce the quantization error caused by the differences between class distributions, we propose *class imbalanced quantization (ClimbQ)*, as shown in Fig. 2, which scales the class data to obtain closer distributions (see Sec. 3.1.1) and project them to the same space for quantization (see Sec. 3.1.2).

### 3.1.1 Class distribution scaling and generation

According to Fig. 1c, majority classes are always distributed in wide ranges, whereas minority classes are distributed in small variations. Such different class distributions cause massive quantization errors and derive a notable accuracy loss, especially for the minority classes (validated in Fig. 1b). As a result, we aim to diminish majority class variations while enlarging minority class variations. First, we denote the network features for $N$ data as $X = \{x_1, x_2, ..., x_N\}$ and suppose that the data number for the $k$-th class is $n_k$, with $k = 1, 2, ..., K$ (in total $K$ classes). Assume the $k$-th class distribution $f_k$ follows the normal distribution denoted as $N(\mu_k, \sigma_k^2)$, where $\mu_k$ and $\sigma_k$ indicate the mean and standard deviation of the distribution[2]. We scale the $k$-th class distribution $f_k : N(\mu_k, \sigma_k^2)$ to $f_k' : N(\mu_k, (c_k \sigma_k)^2)$, where $c_k$ is the *scale factor* defined as $2 - \frac{n_k}{\sum_{i=1}^{K} n_i}, \forall k = 1, 2, ..., K$[3]. The design reveals that the (minority) classes with less data have been scaled to larger variances. The variances of the (majority) classes with a large amount of data are reduced. In the next subsection, we will develop an efficient quantization method based on the scaled class distributions.

### 3.1.2 Class distribution quantization

As illustrated in Sec. 3.1.1, data are scaled to $N(\mu_k, (c_k \sigma_k)^2)$ based on the class it belongs to. In the following, we aim to design an efficient quantization process based on the new class distributions. Recent works adopted and efficient process, the uniform quantization, due to its simplicity and hardware-friendliness in operations on devices [6, 5][4]. However, since data distributions are typically not uniformly distributed, the quantization process produces massive errors [34, 35, 7, 5]. Prior research therefore proposed to learn a clipping function, quantization statistics, and other preprocessing procedures to reduce the error [7, 5, 11, 16]. Nevertheless, the proposed preprocessing procedures are complicated and necessarily require additional learning parameters. Therefore, we propose a one-step transformation without additional learning parameters that allows the class data to be projected to a uniform space with a minimal error under the uniform quantization scheme.

---

[2]Previous research have studied that CNN weights and features follow normal distributions [6, 5]. In addition, we validate the normality on the benchmark datasets in Appendix D. Therefore, in this study, we assume data follow the normal distributions.

[3]We adopt the distribution function of the standard normal $N(0, 1)$, instead of $N(\mu_k, (c_k \sigma_k)^2)$, for inference, since the class information is unknown for the testing data. Note that the main reason that we utilized the standard normal is that the features had been normalized through the batch normalization layer before quantization.

[4]A uniform quantization scheme quantizes the continuous data values to the discrete values with the same intervals.

We adopt the *distribution function* (namely, *cumulative distribution function*) as the uniform space projection function. In the following theorem according to [36], we prove that the data are uniformly distributed after the projection.

**Theorem 3.1.** *(Proved in Appendix A.1) Let $f$ be the continuous probability function of random variable $X$ with its distribution function $F$, then $F(X) = U$, where $U$ is the random variable of the standard uniform distribution.*

Therefore, by Theorem 3.1, we exploit the distribution function of the new class distributions $f_k' : N(\mu_k, (c_k\sigma_k)^2), \forall k$ (determined in Sec. 3.1.1), to project data $x_k$ from the $k$-th class to the uniform space denoted as $D_k$. The projection function is then formulated as:

$$D_k(x_k) = \int_{-\infty}^{x_k} \frac{1}{c_k\sigma_k\sqrt{2\pi}} e^{-\frac{(t-\mu_k)^2}{2c_k^2\sigma_k^2}} \, dt, \ \forall k = 1, 2, ..., K, \tag{1}$$

where $c_k$ is the scale factor introduced and defined in Sec. 3.1.1. Eq. (1) indicates that the data projection is conditioned on its belonged class and the data size. After transformed to the uniform space, the data $x_k' := D_k(x_k)$ is then processed by the following uniform quantization function:

$$Q(x_k') = \frac{round(2^{B-1} \cdot x_k')}{2^{B-1}}, \tag{2}$$

where $round$ is the rounding operation, and $B$ represents the quantization bits.

In summary, as depicted in Fig. 2a, the data from different classes are scaled to diminish the differences between class distributions (detailed in Sec. 3.1.1) and then projected to the uniform space for quantization (see Eq. (1) and Eq. (2)). ClimbQ process is applied in each convolutional layer as presented as the red module in Fig. 2c, i.e., the quantized data $Q(x_k')$ output from the last layer acts as the input $x_k$ for the current layer. The effectiveness of the scaling on quantization error reduction is demonstrated in Appendix E.

## 3.2 Rebalancing for homogeneity of variances

In Sec. 3.1, we proposed distribution scaling and uniform space projection to reduce the differences between class distributions for a minimal quantization error. To ensure homogeneity of the class distributions after quantized through the ClimbQ process, in this subsection, we examine the variances of the quantized features from different classes based on *Levene*'s hypothesis testing. Our main contribution is the derivation of the bounds of the data size for each class to satisfy the homogeneity (illustrated in the blue modules in Fig. 2b and Fig. 2c). Accordingly, we design a *HomoVar* loss that focuses on the classes with data sizes out of the bounds (as illustrated in the green modules in Fig. 2b and Fig. 2c). To prevent the results from being dominated by the majority classes [29, 30, 31, 32], we intend to weight down the losses of those classes whose data size exceeds the upper bound, i.e., the majority classes, while putting more weights on the losses of the minority classes with a smaller data size than the lower bound.

### 3.2.1 Appropriate class data size estimation

For a systematic analysis of the homogeneity of class distributions following the ClimbQ process (introduced in Sec. 3.1), we refer to *Levene*'s hypothesis testing based on the theory of statistical learning [13]. Assume there are total $K$ classes in the dataset. Two hypotheses are included in Levene's testing: *null hypothesis $H_0 : \sigma_1^2 = \sigma_2^2 = ... = \sigma_K^2$* and *alternative hypothesis $H_a : \sigma_i^2 \neq \sigma_j^2$*, for arbitrary $i \neq j$, $i, j = 1, 2, ..., K$. The null hypothesis claims the homogeneity of variances, whereas the alternative hypothesis stands for the heterogeneity if any variance is much different from the others.

According to Levene's hypothesis testing, the null hypothesis is satisfied when the testing statistics $W$ is small. The testing statistics $W$ is formulated as the following:

$$W = \frac{\sum_{k=1}^{K} n_k(\bar{z}_{k\cdot} - \bar{z}_{\cdot\cdot})^2/(k-1)}{\sum_{k=1}^{K} \sum_{i=1}^{n_k} (\bar{z}_{ki} - \bar{z}_{k\cdot})^2/(N-k)} = \frac{S_B}{S_W}, \tag{3}$$

where $z_{ki} = |x_{ki} - \frac{1}{n_k} \sum_{i=1}^{n_k} x_{ki}|$ is the variation, i.e., the distance of data $x_{ki}$ from the center of the $k$-th class[5], $\bar{z}_{k\cdot} = \frac{1}{n_k} \sum_{i=1}^{n_k} z_{ki}$ represents the mean variation of the $k$-th class, and $\bar{z}_{\cdot\cdot} = \frac{1}{K} \sum_{k=1}^{K} \bar{z}_{k\cdot}$

---

[5] $x$ here is as same as the notation in Sec. 3.1 that denotes the data (features) of convolutional layers.

stands for the mean variation for all data. In addition, $n_k$ and $N$ individually denote the data size of the $k$-th class and total data size, i.e., $N = \sum_{k=1}^{K} n_k$.

The denominator of $W$ in Eq. (3) represents the sum of the differences from the *within* class variations, whereas the numerator shows the total difference from the *between* class variations. Thus, we denote the former as $S_W$ and the latter as $S_B$ in Eq. (3). The null hypothesis is accepted when $W$ is small, i.e., $W < F_\alpha(k-1, N-k)$, where $F_\alpha(k-1, N-k)$ is the statistics of $F$-distribution with degree of freedoms, $k-1$ and $N-k$ under the *significance level* $\alpha$ that manifests the statistical significance on the testing results [6].

Derived from $W < F_\alpha(k-1, N-k)$, we prove in the following theorem that each class data size is restricted with a lower bound $n_k^L$ and an upper bound $n_k^U$, $\forall k = 1, 2, ..., K$. It indicates that when class data sizes are not too tiny or excessively large, class variances are homogeneous ($H_0$ is satisfied). On the other hand, when class data sizes are too small or large, significant differences appear between class variances ($H_0$ is rejected).

**Theorem 3.2.** *(Proved in Appendix A.2) Given the definitions and the notations in Eq. (3), if $W < F_\alpha(k-1, N-k)$, where $F_\alpha(k-1, N-k)$ is the statistics of F-distribution with degree of freedoms $(k-1, N-k)$ under the significance level $\alpha$, then class data size $n_k^L < n_k < n_k^U$, $\forall k = 1, 2, ..., K$. The bounds $n_k^L, n_k^U = \frac{2\bar{z}_{..} \cdot \sum_{i=1}^{n_k} z_{ki} + C \pm \sqrt{(2\bar{z}_{..} \cdot \sum_{i=1}^{n_k} z_{ki} + C)^2 - 4\bar{z}_{..}^2 \cdot (\sum_{i=1}^{n_k} z_{ki})^2}}{2\bar{z}_{..}^2}$, where $C = (k-1) \cdot F_\alpha(k-1, N-k) \cdot S_W - \sum_{j \neq k}^{K} n_j (\bar{z}_{j.} - \bar{z}_{..})^2$.*

Theorem 3.2 demonstrates that if the data size of class $n_k$ is in the interval $[n_k^L, n_k^U]$, $H_0$ is satisfies, i.e., the class variances are homogeneous. Otherwise, $H_0$ is rejected, which reveals that classes with out-of-range data sizes (too minor or too large) may violate the homogeneity of class variances. Therefore, in the subsequent subsection, we design a loss using the analyzed bounds to reweight the data losses of each classes to adapt the class distributions to satisfy the homogeneity.

### 3.2.2 Loss for homogeneity of class variances

In the preceding subsection, we derived the bounds of data sizes for each class that satisfies the homogeneity of class variances (see Eq. (3.2)). Consequently, in this subsection, we propose a reweighted loss, named *HomoVar loss*, to reweight the classes that may exhibit significant differences from other classes. We consider three cases on classes: 1) **minority classes** with a data size below the lower bound, 2) **moderated classes** with a data size within the required range, and 3) **majority classes** with a data size exceeding the upper bound.

To avoid the majority from dominating the prediction outcomes, we primarily focus on the minority and downweight the majority (shown in Fig. 1b). That is, the minority classes are penalized the greatest, followed by the middle and then the majority. In addition, we evaluate the degree of minority or majority of a class, i.e. how close the class data size is to the nearest bound. Classes with a minor (enormous) data size far from the derived bounds are weighted more heavily (weighted less).

According to the above insights, we design the HomoVar loss denoted as $\mathscr{L}_H$ in the following:

$$\mathscr{L}_H = \frac{1}{N} \sum_{k=1}^{K} (\omega_k \cdot \sum_{i=1}^{n_k} \mathscr{L}_{ki}) = \frac{1}{N} \sum_{k=1}^{K} (\frac{1 - \beta^{\frac{1}{|n_k - n_k^e|}}}{1 - \beta^{\frac{n_k}{|n_k - n_k^e|}}} \cdot \sum_{i=1}^{n_k} \mathscr{L}_{ki}), \tag{4}$$

where $\mathscr{L}_{ki}$ represents the loss on the $i$-th data of the $k$-th class, $\omega_k = \frac{1 - \beta^{\frac{1}{|n_k - n_k^e|}}}{1 - \beta^{\frac{n_k}{|n_k - n_k^e|}}}$ is the weight on the data of the $k$-th class, $\beta$ is a constant factor in $(0, 1)$ that scales the losses on different classes (discussed in Appendix B). The term $|n_k - n_k^e|$ is used to measure the degree of the minority and majority, where $n_k^e$, namely the expected data size, is defined as the nearest bound to the data size if the actual data size $n_k$ is out of the range $[n_k^L, n_k^U]$, otherwise, $n_k - 1$ [7].

When the $k$-th class is an extraordinarily minor class, the weight $\omega_k$ is around 1. Moreover, $\omega_k$ is near to $\frac{1}{\sum_{i=0}^{n_k - 1} \beta^i} \leq 1$ when $n_k$ is inside the restricted range satisfying the homogeneity of class

---

[6]A larger $\alpha$ results in a smaller $F_\alpha(k-1, N-k)$ which is of greater evidence to support the homogeneity of class differences. In statistical hypothesis testing, the significance level $alpha$ is typically set to 0.1, 0.05, or 0.001. In this paper, we adopt the significance level of 0.05.

[7]We set $n_k^e$ to $n_k - 1$ instead of $n_k$ is to avoid the zero denominator in Eq. (4).

Table 1: Accuracy (%) on CIFAR-10-LT under quantization. * indicates the quantization approach fails at the imbalance ratio $\gamma$.

| Methods | 4-bit ResNet-20 | | | 2-bit ResNet-20 | | | 4-bit MobileNet-V2 | | | 2-bit MobileNet-V2 | | |
|---|---|---|---|---|---|---|---|---|---|---|---|---|
| | $\gamma = 10$ | 50 | 200 | $\gamma = 10$ | 50 | 200 | $\gamma = 10$ | 50 | 200 | $\gamma = 10$ | 50 | 200 |
| LLSQ [15] | 68.57 | 48.20 | 58.90 | * | * | * | 64.28 | 53.08 | 34.08 | * | * | * |
| ZeroQ [18] | 79.37 | 69.40 | 58.70 | 77.75 | 69.25 | 59.10 | 80.05 | 69.85 | 58.93 | * | * | * |
| Choi *et al.* [17] | 79.69 | 69.74 | 59.42 | 77.80 | 69.36 | 58.46 | 80.13 | 69.98 | 61.04 | 59.25 | * | * |
| ZAQ [10] | 79.81 | 69.71 | 59.98 | 77.72 | 70.03 | 58.79 | 80.15 | 70.82 | 58.49 | * | * | * |
| Qimera [11] | 79.22 | 69.96 | 61.57 | 53.85 | 39.82 | 34.16 | * | * | * | * | * | * |
| BatchQuant [16] | 76.98 | 69.19 | 56.55 | 71.74 | 59.39 | 54.27 | 73.93 | * | * | 16.24 | * | * |
| **ClimbQ** | **80.72** | 70.90 | 61.31 | **79.25** | 70.33 | 59.44 | 78.91 | 71.77 | 60.98 | 69.94 | 61.74 | 52.47 |
| **ClimbQ+** | 80.58 | **72.28** | **61.69** | 78.88 | **71.73** | **61.06** | **80.24** | **71.82** | **62.10** | **72.51** | **63.41** | **56.89** |

variances. Additionally, it demonstrates that $\frac{1}{\sum_{i=0}^{n_k-1} \beta^i}$ decreases as the size $n_k$ increases, eventually approaching $1 - \beta$ when $n_k$ is exceedingly large, i.e., a majority class. If the data size is far beyond the upper bound, $\omega_k$ approaches zero. The results are consistent with the ideas behind the design: 1) the weights of minority classes are heavier than those of majority classes, 2) the weights increase as the class data size falls far below the lower bound, and 3) the weights reduce as the class data size exceeds the upper bound. We analyze and discuss further details in Appendix A.3.

# 4 Experiments

## 4.1 Experiment settings

**Class-imbalanced datasets.** We evaluate the effectiveness of ClimbQ and ClimbQ+ (ClimbQ with HomoVar loss) on the class-imbalanced datasets, Syndigit-LT [37], CIFAR-10-LT [38] and CIFAR-100-LT [38]. A parameter setting $\gamma$ determines the degree of imbalance in the datasets. $\gamma$ is the ratio between the data size of the maximum class and the size of the minimum class. For example, $\gamma = 100$ indicates that the data size of the largest class is 100 times larger than the minimum class. In this paper, we train at imbalance ratios of 10, 50, and 200 and validate on the balanced testing data (i.e., $\gamma = 1$) to fairly evaluate the performance of each class.

**Benchmark balanced dataset.** Although we are primarily concerned with imbalanced data, we compare with the baseline quantization approaches on the benchmark dataset ImageNet-ILSVRC 2012 [39]. The imbalanced ratio is 1.77, which is close to perfectly balanced data ($\gamma = 1$).

**Architectures.** We use ResNets [40] and MobileNet-V2 [41] as the underlying architectures. ResNet is a benchmark architecture for quantization research, while MobileNet-V2 is essentially an efficient architecture with lightweight module designs that is also commonly used in image recognition.

**Training.** We utilize an NVIDIA Tesla V100 GPU and an NVIDIA GTX 2080Ti for implementation. ImageNet has a batch size of 512, while Syndigits-LT, CIFAR-10-LT, and CIFAR-100-LT have 128. The maximum number of training epochs is 200. The range of the learning rate is from 0.01 to 0.1. The significance level $\alpha$ referring to Theorem 3.2 is set to 0.05. The constant factor $\beta$ in Eq. (4) is set to 0.999. More discussions about the setting are detailed in Appendix B. Code is available at `https://github.com/tinganchen/ClimbQ.git`.

## 4.2 Comparison results

In the following, we compare the quantization performance of ClimbQ and ClimbQ+ with QAT [8, 15, 11, 16] and ZSQ baselines [19, 18, 17, 10] on the class-imblanced datasets with distinct imbalance ratio settings[8].

**CIFAR-10-LT.** The results of 4-bit and 2-bit quantization of ResNet-20 and MobileNet-V2 on CIFAR-10-LT are presented in Table 1. First, we can observe that the prediction accuracy declines with the imbalance ratio $\gamma$ increases. In addition, the process of low-bit quantization reduces precision.

---

[8]In this paper, model weights and features are quantized to low bits for each convolution layer. The baselines being compared have the same quantization settings.

Table 2: Accuracy (%) on CIFAR-100-LT under quantization. * indicates the quantization approach fails at the imbalance ratio $\gamma$.

| Methods | 4-bit ResNet-56 | | | 2-bit ResNet-56 | | | 4-bit MobileNet-V2 | | | 2-bit MobileNet-V2 | | |
|---|---|---|---|---|---|---|---|---|---|---|---|---|
| | $\gamma = 10$ | 50 | 200 | $\gamma = 10$ | 50 | 200 | $\gamma = 10$ | 50 | 200 | $\gamma = 10$ | 50 | 200 |
| LLSQ [15] | * | * | * | * | * | * | 34.46 | 26.23 | 11.60 | * | * | * |
| ZeroQ [18] | 5.74 | 4.92 | 1.38 | 5.47 | 5.27 | 1.27 | 53.79 | 27.29 | 20.91 | 25.91 | 15.89 | 13.56 |
| Choi *et al.* [17] | 6.25 | 4.09 | * | 5.03 | 1.18 | 1.00 | 53.84 | 36.38 | 28.30 | 26.51 | 15.27 | 12.38 |
| ZAQ [10] | 18.81 | * | * | * | * | * | 52.86 | 35.70 | 27.15 | * | * | * |
| Qimera [11] | 48.64 | 34.44 | * | 8.60 | 2.98 | * | * | * | * | * | * | * |
| BatchQuant [16] | 41.50 | 28.91 | 22.91 | 37.24 | 27.03 | 22.17 | 41.50 | 23.77 | 21.48 | 35.80 | 23.79 | 10.69 |
| **ClimbQ** | 50.04 | 34.30 | 25.64 | 47.15 | 33.85 | 23.91 | 53.74 | 31.47 | 22.58 | 41.50 | 22.02 | 18.53 |
| **ClimbQ+** | **52.13** | **35.10** | **27.46** | **49.12** | **34.79** | **27.39** | **55.62** | **36.89** | **28.47** | **43.96** | **33.18** | **25.45** |

In particular, most of the compared quantization approaches have a significant accuracy loss for low bit models in highly imbalanced cases, since they quantize on imbalanced data (with heterogeneous class distributions) using the same quantization functions or distillation method, which results in a huge quantization error and a significant loss of accuracy. In contrast, our ClimbQ and ClimbQ+ quantization schemes are proposed to diminish the heterogeneity in different classes (introduced in Sec. 3). Table 1 demonstrates that our proposed methodologies can effectively reduce quantization error and accuracy loss, especially on the lightweight architecture design, MobileNet-V2. ClimbQ increases the accuracy of baselines under all studied conditions. In addition, ClimbQ yields a performance increase of up to 10% for 2-bit MobileNet-V2 with $\gamma = 10$ and can successfully implemented on the highly imbalanced cases $\gamma = 50$ and 200. ClimbQ+ can further improve the performance of ClimbQ by as much as 4.4% in absolute accuracy. The exceptional performance is a result of attempts to reduce the heterogeneity of class variances using quantized features and class data sizes (see the evaluation of homogeneity and the proposed HomoVar loss in Sec. 3.2).

**CIFAR-100-LT.** We also study the efficacy of ClimbQ and ClimbQ+ on CIFAR-100-LT, which contains 10 times as many classes as CIFAR-10-LT. Due to the increasing difficulty of the recognition task, we employ a deeper ResNet-56 model instead of ResNet-20 for quantization. Table 2 reveals that the accuracy of the compared approaches degrades significantly under highly imbalanced circumstances. ClimbQ+ enhances the performance of the BatchQuant by 5% to 12% accuracy on 2-bit ResNet-56 and increases 8% to 15% for 2-bit MobileNet-V2, validating the effectiveness of the ClimbQ process (scaling the class variances as introduced in Sec. 3.1) and the HomoVar loss (rebalancing the data size of each class as shown in Sec. 3.2) on quantization error reduction.

**Syndigits-LT and ImageNet.** We further evaluate and discuss the performances of ClimbQ and ClimbQ+ on the imbalanced dataset Syndigits-LT and balanced benchmark ImageNet in Appendix C. ClimbQ+ on Syndigits-LT in the highly imbalanced case, $\gamma = 200$, achieves a remarkable 80.55% accuracy for 2-bit ResNet-20, i.e., a 22% accuracy improvement over the best baseline result evaluated (only 58.75% accuracy). Moreover, ClimbQ+ on 2-bit MobileNet-V2 outperforms the state-of-the-art BatchQuant with a 26% increase in accuracy. In addition to the imbalanced datasets, we compare our quantization methods against those previously proposed on the balanced benchmark ImageNet. ClimbQ+ outperforms the previous studies by approximately 3% in terms of accuracy. More details are discussed in Appendix C.

## 5 Ablation study on rebalancing strategies

In Sec. 4, we demonstrated the superior performance of ClimbQ (introduced in Sec. 3.1) and ClimbQ+ (described in Sec. 3.2) over the baseline quantization approaches, which is particularly with a significant improvement for low-bit and lightweight models in extremely imbalanced scenarios. In this section, we compare ClimbQ+ to other effective rebalancing methods[9]. Table 3 demonstrates that ClimbQ+ has outstanding performances compared to the previous research on rebalancing data sizes. ClimbQ+ improves absolute accuracy by 4.4% to 22.4% for the 2-bit model on CIFAR-10-LT with $\gamma = 50$ and by 2.6% to 25.3% accuracy with $\gamma = 200$. The extraordinary performances of ClimbQ+ verifies that the HomoVar loss introduced in Sec. 3.2 can effectively reduce more quantization errors

---

[9]The rebalancing methodologies are implemented under proposed ClimbQ quantization process on the imbalanced data to compare with our reweighted loss, the HomoVar loss (designed in Sec. 3.2).

Table 3: Accuracy (%) of rebalancing approaches on ClimbQ based low-bit ResNet-20. * indicates the quantization approach fails at the imbalance ratio $\gamma$.

| Methods | Syndigits-LT (4-bit) | | | Syndigits-LT (2-bit) | | | CIFAR-10-LT (4-bit) | | | CIFAR-10-LT (2-bit) | | |
|---|---|---|---|---|---|---|---|---|---|---|---|---|
| | $\gamma = 10$ | 50 | 200 | $\gamma = 10$ | 50 | 200 | $\gamma = 10$ | 50 | 200 | $\gamma = 10$ | 50 | 200 |
| Focal [29] | 96.05 | 88.20 | 76.35 | 95.20 | 87.15 | 76.05 | 80.52 | 66.69 | 55.80 | 78.27 | 55.96 | 37.69 |
| CB [30] | 96.10 | 88.45 | 76.45 | 95.50 | 87.60 | 74.80 | 76.31 | 69.70 | 60.46 | 65.62 | 59.94 | 51.69 |
| LDAM [32] | 95.25 | 86.85 | 78.15 | 94.80 | 86.35 | 76.90 | 80.48 | 70.68 | 56.01 | 78.65 | 49.38 | 37.21 |
| Causal [33] | 96.20 | 88.65 | 79.70 | 95.40 | 87.90 | 78.50 | 76.60 | 69.81 | 60.05 | 73.44 | 67.33 | 58.44 |
| LADE [31] | 95.45 | 85.65 | 72.10 | 95.15 | 85.45 | 71.80 | **80.69** | 64.99 | 49.56 | 78.88 | 60.24 | 47.45 |
| **ClimbQ+** | **96.45** | **89.55** | **80.70** | **95.75** | **88.90** | **80.55** | 80.58 | **72.28** | **61.69** | **78.88** | **71.73** | **61.06** |

and accuracy loss owing to the rebalancing strategy that takes the homogeneity of the class variances (variances of quantized features) into a consideration, rather than only leveraging the class data sizes.

# 6 Discussion with potential impacts and limitations

**Potential impacts.** ClimbQ with the HomoVar loss aims to rebalance the classes for homogeneity of class variances. To prevent the majority classes from dominating the quantization results, the rebalancing strategy focuses primarily on the minority classes while scaling down the weights on the majority classes for the homogeneity of class variances, which implies that the prediction results of the minority classes are more dependent on the small amount of data. Accordingly, they may be much easier to be attacked with a notable performance degradation. However, recent research has developed the defense strategy against the attacks on imbalanced data [42] which can also be applied to ClimbQ.

**Limitations.** ClimbQ mainly focuses on the class-imbalanced data with different class data sizes and distributions. However, we believe that there are other scenarios of heterogeneous data in the real world that have not been fully studied. As a result, we will explore and study more research issues of quantization on heterogeneous data in our future works.

# 7 Conclusion

In this paper, we study a realistic but not fully studied research problem of quantization on class-imbalanced data, where the differences in class distributions lead to a huge quantization error in the minority classes. To address this issue, we propose ClimbQ to scale the class distributions to diminish the differences between classes for quantization error reduction. Moreover, we analyze that the homogeneity of class variances under ClimbQ process is subjected to restricted class data sizes. Therefore, we further propose HomoVar loss to rebalance the classes based on the derived bounds of data sizes to satisfy the homogeneity. Experiments reveal that ClimbQ/ClimbQ+ outperforms the baseline quantization and rebalance techniques, validating the effectiveness on quantization error reduction by diminishing the differences in class distributions.

# Acknowledgment

This work was supported in part by the Ministry of Science and Technology, Taiwan, under grant MOST 111-2223-E-002 -006 and MOST 111-2221-E-002 -135 -MY3.

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
