# Supplementary Materials of ClimbQ: Class Imbalanced Quantization Enabling Robustness on Efficient Inferences

**Ting-An Chen**[1,2]**, De-Nian Yang**[2,3] **Ming-Syan Chen**[1,3]
[1]Graduate Institute of Electrical Engineering, National Taiwan University, Taiwan
[2]Institute of Information Science, Academia Sinica, Taiwan
[3]Research Center for Information Technology Innovation, Academia Sinica, Taiwan
`tachen@arbor.ee.ntu.edu.tw, dnyang@iis.sinica.edu.tw, mschen@ntu.edu.tw`

## A. Proof and theoretical analysis

### A.1 Distribution function for class distribution generation

After the determination of scaled class distributions in Sec. 3.1.1 of the main paper, we target designing an efficient quantization process. Since quantization on he uniform space is computationally efficient and therefore hardware-friendly for deployment on devices, we aim to project data to the uniform space prior to quantization. Inspired by the property in probability distributions that the distribution function of any continuous probability distribution follows a standard uniform distribution, we adopt the distribution function as the data projection before the quantization. The following demonstrates the property:

**Theorem 3.1.** *Let $f$ be the continuous probability function of random variable $X$ with its distribution function $F$, then $F(X) = U$, where $U$ is the random variable of the standard uniform distribution.*

*Proof.* Distribution function with definition in the range of [0, 1]. The distribution function of X is
$$P(X \leq x) = P[F(X) \leq x] = P[X \leq F^{-1}(x)]$$
$$= F[F^{-1}(x)] = x, \ 0 \leq x \leq 1,$$
which is the distribution function of a Uniform(0, 1) (standard uniform) random variable. $\square$

Theorem 3.1. proves that the distribution function can projects data to the uniform space, which is favorable in efficient uniform quantization with simple operations.

### A.2 Appropriate class data size estimation

To reduce quantization error in imbalanced class distributions, we aim to diminish the discrepancies between class distributions. Therefore, we propose a distribution scaling on class variances in Sec. 3.1.1 of the main paper. To ensure the homogeneity of the variances, we examine it in accordance with Levene's hypothesis in Sec. 3.1.2. Moreover, we derive from the analytical results that the homogeneity criterion is satisfied if the data size of each class is restricted. In the following, we estimate the bounds with a statistical significance $\alpha$.

**Theorem 3.2.** *Given the definitions and the notations in Eq. (3) and its subsequent paragraphs of the main paper, if $W < F_\alpha(k-1, N-k)$, where $F_\alpha(k-1, N-k)$ is the statistics of F-distribution with degree of freedoms $(k-1, N-k)$ under the significance level $\alpha$, then class data size $n_k^L < n_k < n_k^U, \forall k = 1, 2, ..., K$. The bounds $n_k^L, n_k^U = \frac{2\bar{z}_{..} \cdot \sum_{i=1}^{n_k} z_{ki} + C \pm \sqrt{(2\bar{z}_{..} \cdot \sum_{i=1}^{n_k} z_{ki} + C)^2 - 4\bar{z}_{..}^2 (\sum_{i=1}^{n_k} z_{ki})^2}}{2\bar{z}_{..}^2}$, where $C = (k-1) \cdot F_\alpha(k-1, N-k) \cdot S_W - \sum_{j \neq k}^{K} n_j (\bar{z}_{j\cdot} - \bar{z}_{..})^2$.*

36th Conference on Neural Information Processing Systems (NeurIPS 2022).

*Proof.* Since

$$W = \frac{\sum_{k=1}^{K} n_k (\bar{z}_{k\cdot} - \bar{z}_{\cdot\cdot})^2/(k-1)}{\sum_{k=1}^{K}\sum_{i=1}^{n_k}(\bar{z}_{ki} - \bar{z}_{k\cdot})^2/(N-k)} = \frac{S_B}{S_W} < F_\alpha(k-1, N-k),$$

which implies that

$$\sum_{k=1}^{K} n_k(\bar{z}_{k\cdot} - \bar{z}_{\cdot\cdot})^2 < (k-1)\cdot F_\alpha(k-1, N-k)\cdot S_W$$

$$\implies n_k(\bar{z}_{k\cdot} - \bar{z}_{\cdot\cdot})^2 + \sum_{j \neq k}^{K} n_j(\bar{z}_{j\cdot} - \bar{z}_{\cdot\cdot})^2 < (k-1)\cdot F_\alpha(k-1, N-k)\cdot S_W$$

$$\implies n_k(\bar{z}_{k\cdot} - \bar{z}_{\cdot\cdot})^2 < (k-1)\cdot F_\alpha(k-1, N-k)\cdot S_W + \sum_{j \neq k}^{K} n_j(\bar{z}_{j\cdot} - \bar{z}_{\cdot\cdot})^2$$

Let the right handside of the above inequality be a constant $C$[1]. Therefore,

$$n_k(\bar{z}_{k\cdot} - \bar{z}_{\cdot\cdot})^2 < C.$$

By expanding the left handside, we obtain that

$$n_k\left(\frac{\sum_{i=1}^{n_k} z_{ki}}{n_k} - \bar{z}_{\cdot\cdot}\right)^2 < C,$$

$$\implies \frac{(\sum_{i=1}^{n_k} z_{ki})^2}{n_k} - 2\cdot\left(\sum_{i=1}^{n_k} z_{ki}\right)\cdot\bar{z}_{\cdot\cdot} + n_k\cdot\bar{z}_{\cdot\cdot}^2 < C$$

$$\implies \bar{z}_{\cdot\cdot}^2\cdot n_k^2 - \left(2\bar{z}_{\cdot\cdot}\cdot\sum_{i=1}^{n_k} z_{ki} - C\right)\cdot n_k + \left(\sum_{i=1}^{n_k} z_{ki}\right)^2 < 0.$$

Thus,

$$n_k > \frac{2\bar{z}_{\cdot\cdot}\cdot\sum_{i=1}^{n_k} z_{ki} + C - \sqrt{(2\bar{z}_{\cdot\cdot}\cdot\sum_{i=1}^{n_k} z_{ki} + C)^2 - 4\bar{z}_{\cdot\cdot}^2(\sum_{i=1}^{n_k} z_{ki})^2}}{2\bar{z}_{\cdot\cdot}^2}$$

and

$$n_k < \frac{2\bar{z}_{\cdot\cdot}\cdot\sum_{i=1}^{n_k} z_{ki} + C + \sqrt{(2\bar{z}_{\cdot\cdot}\cdot\sum_{i=1}^{n_k} z_{ki} + C)^2 - 4\bar{z}_{\cdot\cdot}^2(\sum_{i=1}^{n_k} z_{ki})^2}}{2\bar{z}_{\cdot\cdot}^2}.$$

□

Hence, we obtain the bounds of data size for each class to satisfy the homogeneity of class variances (Levene's hypothesis) with statistical significance. According to the result, we design a loss to reweight the class data in the next subsection (see Sec. 3.2.2 of the main paper).

### A.3. Analysis and discussion on HomoVar loss

The HomoVar loss is proposed in Sec. 3.2.2, Eq. (4) in the main paper. The idea behind is to reweight the loss on classes for the homogeneity of class variances, which addresses the problem of quantization errors resulting from different class distributions. The weight on the $k$-th class $\omega_k$ is defined as $\frac{1 - \beta^{\frac{1}{|n_k - n_k^e|}}}{1 - \beta^{\frac{n_k}{|n_k - n_k^e|}}}$ in Sec. 3.2.2 of the main paper and also illustrated in Table 1, where we adopt the constant factor $\beta \in (0, 1)$. In this section, we discuss with the designs through a series of systematic analyses.

---

[1]We regard the term including $S_W$ as a constant is because the $S_W$'s for different classes have been scaled closer. In particular, the $S_W$'s of minority classes are generally small but enlarged (see Sec. 3.1.1)

Table 1: Class weight based on the approximation of actual class data size to expected size.

| Cases | Case I. $|n_k - n_k^e| \to 1$ Approximate homogeneity of variances | Case II. $|n_k - n_k^e| \to \infty$ Heterogeneity of class variances |
|---|---|---|
| Loss weight | $\omega_k = \frac{1}{\sum_{i=0}^{n_k-1} \beta^i}$ | $\omega_k = \frac{1}{n_k}$ |

Table 2: Class weight based on the actual class data size.

| Cases | Case I. $n_k \to 1^+$ Extreme minority class | Case II. $n_k \sim n_k^e$ Moderated sized class | Case III. $n_k^e < n_k \to \infty$ Extreme majority class |
|---|---|---|---|
| Loss weight | $\omega_k = 1$ | $\omega_k = \frac{1}{\sum_{i=0}^{n_k-1} \beta^i}$ | $\omega_k = 0$ |

### A.3.1 Analysis of class weight based on the actual and expected class data sizes

Table 1 shows that $\omega_k = \frac{1}{\sum_{i=0}^{n_k-1} \beta^i}$ when Case I that the actual class data size $n_k$ is close to the expected size $n_k^e$ (defined in Eq. (5) of the main paper, which depends on $n_k$ and the bounds derived from Theorem 3.2). In contrast, $\omega_k = \frac{1}{n_k}$ is calculated in Case II when the actual size is considerably different from the expected size, i.e., when the class size is too small or too large to meet the homogeneity of class variances. The results of these two cases are proved as follows:

*Proof.*

$$\lim_{|n_k - n_k^e| \to 1} \omega_k = \lim_{|n_k - n_k^e| \to 1} \frac{1-\beta}{1-\beta^{n_k}} = \frac{1-\beta}{1-\beta^{n_k}} = \frac{1}{\sum_{i=0}^{n_k-1} \beta^i},$$

and

$$\lim_{|n_k - n_k^e| \to \infty} \omega_k = \lim_{x := \beta^{\frac{1}{|n_k - n_k^e|}} \to 1} \frac{1-x}{1-x^{n_k}} = \lim_{x \to 1} \frac{1}{\sum_{i=0}^{n_k-1} x^i} = \frac{1}{n_k}.$$

$\square$

The results will be utilized in the subsequent section to assess the minority and majority class weights in greater depth.

### A.3.2 Class weight analysis for minority and majority classes

In Appendix A.3.1, we examine the class weight when the class data size is close to or far from the size expected to guarantee the homogeneity of class variances. In this subsection, we discuss the class weight according to whether the class belongs to the minority or the majority. The results of the analysis are displayed in Table 2.

First, when a class belongs to the minority classes, i.e., a class with small $n_k$, the weight $\omega_k$ is near to 1, as shown in Table 1, which can be derived from:

*Proof.*

$$\lim_{n_k \to 1^+} \lim_{|n_k - n_k^e| \to 1} \omega_k = \lim_{n_k \to 1^+} \frac{1}{\sum_{i=0}^{n_k-1} \beta^i} = 1,$$

and

$$\lim_{n_k \to 1^+} \lim_{|n_k - n_k^e| \to \infty} \omega_k = \lim_{n_k \to 1^+} \frac{1}{n_k} = 1.$$

$\square$

The result indicates that the weight on a class with an extremely minor data size is close to one. In addition, the weight is $\frac{1}{\sum_{i=0}^{n_k-1} \beta^i}$ when class data size $n_k$ is close to the expected size, i.e., when the homogeneity of class variances is satisfied, as demonstrated explicitly by Case I. in Table 1.

Moreover, when a class has a huge quantity of data, the weight $\omega_k$ is designed to be significantly less than 1. The case where when $n_k \to \infty$, the weight is close to zero, as demonstrated below:

*Proof.*

$$\lim_{n_k \to \infty} \omega_k = \lim_{n_k \to \infty} \lim_{|n_k - n_k^e| \to 1} \omega_k = \lim_{n_k \to \infty} \frac{1}{\sum_{i=0}^{n_k-1} \beta^i} = 1 - \beta,$$

and

$$\lim_{n_k \to \infty} \omega_k = \lim_{n_k \to \infty} \lim_{|n_k - n_k^e| \to \infty} \omega_k = \lim_{n_k \to \infty} \frac{1}{n_k} = 0.$$

$\square$

Since $\max(1 - \beta, 0) \leq \frac{1}{\sum_{i=0}^{n_k-1} \beta^i} \leq 1, \forall n_k \geq 1$, and $\beta \in (0, 1)$, it is evident that our designs place a greater weight on the minority classes and a smaller weight on the majority classes in order to rebalance the imbalanced classes.

The motivation on how $w_k$ is designed has been described in Sec. 3.2.2 that: 1) the weights of minority classes are heavier than those of majority classes, 2) the weights increase as the class data size falls far below the lower bound, and 3) the weights reduce as the class data size exceeds the upper bound. From the analyses in Appendix A.3.1 and A.3.2, we can observe the approximations of the numerator and denominator. The numerator is designed smaller than the denominator to regularize the $w_k$ ranging in [0, 1]. The growth rate of the denominator is faster when the class size $n_k$ is larger, which leads to the motivation 1). In addition, when the actual class size $n_k$ is much smaller than expected size $n_k^e$, both the numerator and denominator approximate to one, which leads to a larger $w_k$ (i.e., $w_k = 1$). This is consistent with the motivation 2). Moreover, when the $n_k$ is much larger than $n_k^e$, the $w_k$ is smaller than one. The result reflects the motivation 3).

## B. Ablation study of hyperparameter settings

### B.1 Visualization of the class weights based on the different settings of the constant factor

Based on definition of the class weight $\omega_k$ in Sec. 3.2.2 of the main paper, the factor $beta$ determines the weight scaling. Therefore, we will explore the impact of $beta$ on the class weight $\omega_k$. As illustrated in Table 1, the approximate weight is $\frac{1}{n_k}$ regardless of $\beta$ when the actual class data size is much larger or smaller than the expected size. In spite of this, it approximates $\frac{1}{\sum_{i=0}^{n_k-1} \beta^i}$ when the actual size is near to the expected size. Consequently, we primarily examine $\omega_k = \frac{1}{\sum_{i=0}^{n_k-1} \beta^i}$, as visualized in Fig. 1 under distinct configurations of the factor $\beta$ and the data size $n_k$. Fig. 1a illustrates the weights for various $\beta$ values. It is consistent with the analysis results shown in Table 2 and Appendix A.3.2 that 1) the weight lies within the interval [0, 1], 2) the weight declines with increasing class data size, and 3) the weight approaches $1 - \beta$ when $n_k$ is large. Additional observation is that the weights are scaled in different ways based on $\beta$. In Fig. 1b and Fig. 1c, we investigate in depth the minority classes (smaller data sizes) and majority classes (larger data sizes), respectively. We can observe that the larger $\beta$ results in smaller weights for majority classes, i.e., emphasize minority classes more.

### B.2 Performance of ClimbQ+ with various constant factor $\beta$ settings

In Appendix B.1, we illustrated the class weights with various $\beta$ constant factors. Fig. 1 shows that the weights on the majority classes fall as $\beta$ increases. With a large $\beta$, minority classes get more attention than majority classes. To evaluate the effectiveness of $\beta$ in the HomoVar loss, we conduct experiments on different settings of $\beta$ as shown in Table 3. The experimental results reveals that the setting of $\beta = 0.999$ obtains the best performance in all circumstances, validating ClimbQ+ that focuses more on the minority classes can effectively reduce the quantization bias toward the majority classes. Thus, as illustrated in Sec. 4, we adopt $\beta = 0.999$ in this paper.

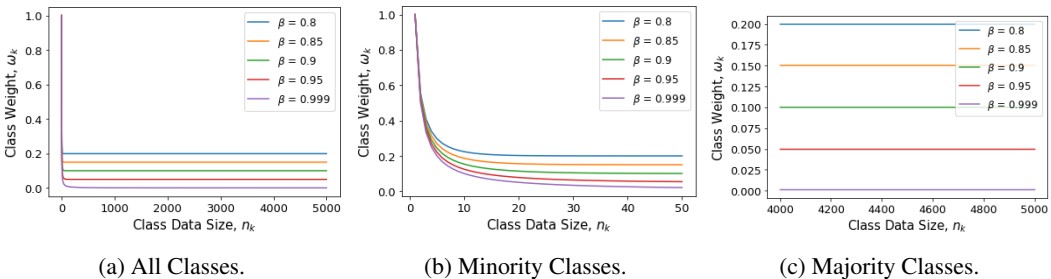

| (a) All Classes. | (b) Minority Classes. | (c) Majority Classes. |

Figure 1: Visualization of class weights in HomoVar loss design based on data size and the settings of constant factor $\beta$.

Table 3: Accuracy (%) of ClimbQ+ on class-imbalanced datasets under the settings of rebalance factor $\beta$.

| Parameters | ResNet-20 on CIFAR-10-LT | | | | ResNet-20 on Syndigits-LT | | | | MobileNet-V2 on Syndigits-LT | | | |
| | 4 bits | | 2 bits | | 4 bits | | 2 bits | | 4 bits | | 2 bits | |
| | $\gamma = 50$ | 200 | $\gamma = 50$ | 200 | $\gamma = 50$ | 200 | $\gamma = 50$ | 200 | $\gamma = 50$ | 200 | $\gamma = 50$ | 200 |
|---|---|---|---|---|---|---|---|---|---|---|---|---|
| $\beta = 0.800$ | 70.43 | 60.33 | 68.68 | 58.15 | 87.65 | 75.85 | 86.30 | 75.35 | 82.60 | 65.25 | 80.85 | 64.55 |
| $\beta = 0.850$ | 71.51 | 60.30 | 68.52 | 57.61 | 87.80 | 76.35 | 86.75 | 75.25 | 81.40 | 65.60 | 80.00 | 64.05 |
| $\beta = 0.900$ | 71.57 | 60.01 | 69.60 | 57.84 | 87.80 | 76.60 | 86.35 | 75.75 | 82.55 | 64.75 | 80.95 | 64.60 |
| $\beta = 0.950$ | 70.85 | 59.75 | 69.58 | 58.99 | 88.20 | 77.45 | 86.75 | 76.15 | 82.40 | 64.95 | 80.75 | 64.65 |
| $\beta = \mathbf{0.999}$ | **72.28** | **61.69** | **71.73** | **61.06** | **89.55** | **80.70** | **88.90** | **80.55** | **83.50** | **66.90** | **83.20** | **67.20** |

## C. Complementary experiments

### C.1. Performance of quantized ResNet-20 and MobileNet-V2 on Syndigits-LT

In Sec. 4.2 of the main study, we assessed ClimbQ and ClimbQ+ on the CIFAR-10-LT and CIFAR-100-LT datasets to demonstrate that they may effectively reduce the quantization error caused by the heterogeneity of class distributions. Here we further compare the baseline quantization approaches on Syndigit-LT as shown in Table 4. It manifests that the previous works quantized at 2 bits obtain a significant accuracy loss compared to 4 bits. In addition, the accuracy loss is in particular large in extremely imbalanced cases. ClimbQ+ outperforms all benchmarks across all evaluation parameters. ClimbQ+ based 2-bit ResNet-20 on $\gamma = 200$ achieves accuracy of 80.55%, outperforming the precious best result (58.75% by Choi *et al.*'s work) with more than 20% accuracy increment. As a meanwhile, ClimbQ+ on $\gamma = 50$ for 2-bit MobileNet-V2 obtains 83.20% accuracy compared to BatchQuant with 57.3% which improves 25% accuracy. The remarkable improvements in accuracy can be attributed to the concept underlying ClimbQ+, which aims to diminish the differences between the class distributions prior to quantization in order to reduce quantization errors. It is validated in Table 4 that ClimbQ+ can achieve a minimal loss of accuracy, particularly for the extremely imbalanced and efficient (low bit) instances.

### C.2. Performance of quantized ResNets on ImageNet

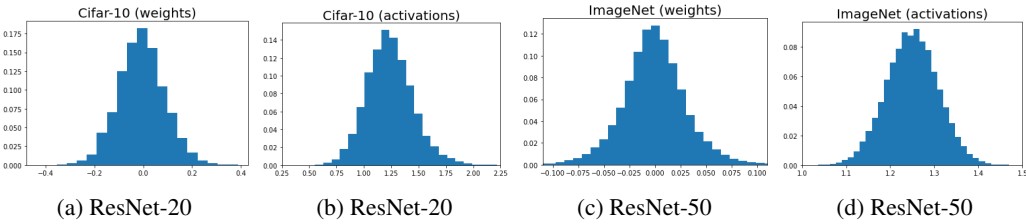

| (a) ResNet-20 | (b) ResNet-20 | (c) ResNet-50 | (d) ResNet-50 |

Figure 2: Distributions of CNN weights and activations on the benchmark datasets, including ResNet-20 on CIFAR-10 and ResNet-50 on ImageNet.

Table 4: Accuracy (%) on Syndigits-LT under quantization. * indicates the quantization approach fails at the imbalance ratio $\gamma$.

| Methods | 4-bit ResNet-20 | | | 2-bit ResNet-20 | | | 4-bit MobileNet-V2 | | | 2-bit MobileNet-V2 | | |
|---|---|---|---|---|---|---|---|---|---|---|---|---|
| | $\gamma = 10$ | 50 | 200 | $\gamma = 10$ | 50 | 200 | $\gamma = 10$ | 50 | 200 | $\gamma = 10$ | 50 | 200 |
| LLSQ [1] | 96.00 | 84.20 | * | * | * | * | 93.25 | 40.60 | * | * | * | * |
| ZeroQ [2] | 95.95 | 87.40 | 73.55 | 93.30 | 75.00 | 58.05 | 90.10 | 63.35 | 42.55 | * | * | * |
| Choi *et al.* [3] | 95.80 | 88.00 | 76.55 | 92.70 | 74.00 | 58.75 | 77.45 | 61.15 | 45.75 | * | * | * |
| ZAQ [4] | 94.75 | 85.75 | 64.50 | * | * | * | 95.30 | 83.30 | 63.45 | 87.25 | 18.25 | * |
| Qimera [5] | 94.00 | * | * | * | * | * | * | * | * | * | * | * |
| BatchQuant [6] | 66.60 | 57.30 | 47.10 | 72.05 | 59.30 | 52.30 | 69.90 | 58.67 | 52.75 | 66.55 | 57.30 | 49.32 |
| **ClimbQ** | 95.90 | 88.07 | 77.25 | 95.30 | 87.05 | 76.75 | **95.50** | 82.75 | 66.45 | **94.60** | 83.05 | 66.30 |
| **ClimbQ+** | **96.45** | **89.55** | **80.70** | **95.75** | **88.90** | **80.55** | 95.45 | **83.50** | **66.90** | 93.50 | **83.20** | **67.20** |

Table 5: Accuracy (%) of quantized ResNets on ImageNet.

| Methods | ResNet-50 | | ResNet-18 | |
|---|---|---|---|---|
| | 4 bits | 2 bits | 4 bits | 2 bits |
| LLSQ [1] | - | - | - | - |
| ZeroQ [2] | 69.30 | 63.12 | 26.00 | - |
| Choi *et al.* [3] | 69.10 | 63.00 | - | - |
| ZAQ [4] | 70.06 | 65.52 | - | - |
| Qimera [5] | 66.25 | - | 63.84 | - |
| BatchQuant [6] | 67.72 | 62.21 | - | - |
| **ClimbQ/ClimbQ+** | **72.73** | **65.68** | **66.14** | **61.13** |

We have validated the effectiveness of ClimbQ and ClimbQ+ on imbalanced datasets in Sec. 4.2 of the main paper and in Appendix C.1. In this subsection, we compare our results against the ImageNet baselines. Table 5 manifests that even though ImageNet is nearly balanced with an imbalance ratio of 1.77, ClimbQ/ClimbQ+ improves the baselines by 2.7% to 5% accuracy for 4-bit ResNet-50 and by 2.3% to 40% accuracy for 4-bit ResNet-18 footnote Since ClimbQ and ClimbQ+ perform similarly in this almost balanced situation, we only present one result for each evaluation setting in Table 5.. These results demonstrate that ClimbQ/ClimbQ+ are effective not only with imbalanced data, but also with balanced data.

## D. Normality of converged CNN weights and activations

In Sec. 3.1.1 of the main paper, we assume that CNN weights and activations (features) converge to a normal distribution which earlier research has investigated [7, 8, 9]. To evaluate the normality, we conduct experiments on benchmark datasets and converged (pretrained) CNN models. Fig. 2 visualizes the distributions of CNN weights and activations. The results are consistent with the findings of earlier studies [7, 8, 9]. Therefore, normal distributions are appropriate for class distributions.

## E. Effectiveness of scaling on quantization error reduction

In Sec. 3.1, we propose to scale the class distributions for quantization error reduction. In this section, we mainly investigate the effectiveness of the scaling on reduction of quantization errors. First, we measure the quantization error as $\frac{|w_q - w|}{\max(w_q) - \min(w_q)}$. The numerator is the discrepancy between the quantized weight and the original floating-point weight. We further normalize the error by dividing the range of the quantized space. Fig. 3 reveals that when the class distributions are not scaled (*"w/o scaling"*), i.e., the differences in the class distributions are not reduced, we have a larger quantization error and therefore obtain a lower accuracy as presented in Table 6 and Table 7. In contrast, after the class distributions are scaled to the similar variations (*"w/. scaling"*) and projected to the same space before the quantization (see Sec. 3.1), the total quantization error is reduced, hence leading to a higher accuracy.

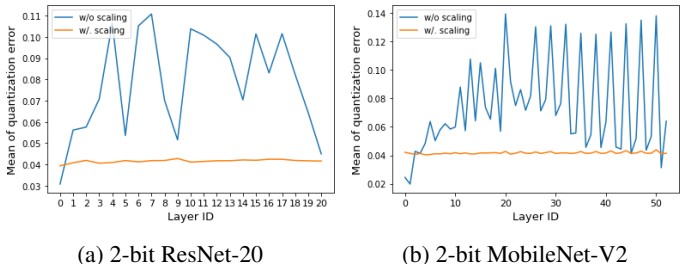

|                    |                    |
| ------------------ | ------------------ |
| (a) 2-bit ResNet-20 | (b) 2-bit MobileNet-V2 |

Figure 3: Comparison of layer-wise quantization errors without/with scaling of class distributions. Fig. 3 (a) presents the result of 2-bit ResNet-20 on CIFAR-10-LT ($\gamma$=50). Fig. 3 (b) shows the result of 2-bit MobileNet-V2 on CIFAR-100-LT ($\gamma$=50).

Table 6: Comparisons of total quantization errors and accuracies (%) without/with scaling of class distributions for 2-bit ResNet-20 on CIFAR-10-LT ($\gamma$=50).

|                            | w/o Scaling | w/. Scaling (Ours) |
| -------------------------- | ----------- | ------------------ |
| Total quantization error   | 21,856      | **11,306**         |
| Test Accuracy (%)          | 68.71       | **70.33**          |

Table 7: Comparisons of total quantization errors and accuracies (%) without/with scaling of class distributions for 2-bit MobileNet-V2 on CIFAR-100-LT ($\gamma$=50).

|                            | w/o Scaling | w/. Scaling (Ours) |
| -------------------------- | ----------- | ------------------ |
| Total quantization error   | 118,317     | **90,810**         |
| Test Accuracy (%)          | 1.07        | **33.18**          |