# OpenReview forum: "ClimbQ: Class Imbalanced Quantization Enabling Robustness on Efficient Inferences"
_NeurIPS.cc/2022/Conference — NeurIPS 2022 Accept_

### Official Review · Reviewer_Q3g7 · 2022-07-10

**Rating:** 7
**Confidence:** 3
**Soundness:** 3 good
**Presentation:** 3 good
**Contribution:** 3 good

**Summary:**

The authors propose ClimbQ quantitzation for efficient inference in
the context of class-imbalanced problems. The majority (minority)
classes have larger (smaller) variance and they reduces (increase) the
variance.  To project onto a uniform distribution, they use a
cumulative distribution function, which is then quantized.  Based on
Levine's hypothesis testing on homogeneity in variances, they show a
lower and upper bound of class data size to reject/accept the null
hypothesis (homogeneous variances).  Using the bounds, they divide the
classes into: minority, moderated, and majority classes.  They then
design the HomoVar loss, which weight minority classes more and
majority classes less.

Empirical evaluation on 3 imbalanced datasets indicates the proposed
method generally outperform 6 existing methods.


**Questions:**

lines 186-188 vs lines 194-195: the null hypothesis is rejected when
the class size is NOT too small/large (lines 186-188), but it is
rejected when the class size is too small/large (lines 194-195).

Eq 4: more motivation on how w_k is designed would be beneficial,
particularly the intent for the numerator and denominator.

line 170:  nominator -> numerator ?


**Limitations:**

Limitations and negative societal impact were discussed.


**Strengths And Weaknesses:**

The problem of quantization for class-imbalanced problem is
interesting.

The proposed projection onto an uniform space and HomoVar loss are
interesting. They derived the upper and lower bounds of class size
with respect to Levine's hypothesis testing on homogeneity in
variances.

Empirical evaluation indicates the proposed approach generally
outperform existing methods.

The paper is generally well written.

---

> ### Author Response · Authors · 2022-08-02
> **Response to Reviewer 4**
>
> We thank the reviewer for taking the time for a review. The replies are listed as the following.
>
>
> Q1. Lines 186-188 vs lines 194-195: the null hypothesis is rejected when the class size is NOT too small/large (lines 186-188), but it is rejected when the class size is too small/large (lines 194-195).
>
> A1.
>
> Actually, what we illustrated in Lines 186-188 is that: ”It indicates that when class data sizes are not too tiny or excessively large, class variances are homogeneous (H0 is satisfied); otherwise, significant differences appear between class variances (H0 is rejected, and Ha is satisfied).”. According to the description, the null hypothesis is rejected when the class size is too small/large.
>
> To avoid the misunderstaing, the referenced content will be rephrased in the revised version as:
> “ It indicates that when class data sizes are not too tiny or excessively large, class variances are homogeneous (H0 is satisfied). On the other hand, when class data sizes are too small or large, significant differences appear between class variances (H0 is rejected).”
>
> Q2. Eq 4: more motivation on how $w_k$ is designed would be beneficial, particularly the intent for the numerator and denominator.
>
> A2.
>
> The motivation on how $w_k$ is designed has been described in Lines 221-224 that: (1) the weights of minority classes are heavier than those of majority classes, (2) the weights increase as the class data size falls far below the lower bound, and (3) the weights reduce as the class data size exceeds the upper bound.
>
> From the analyses in Appendix A.3, we can observe the approximations of the numerator and denominator. The numerator is designed smaller than the denominator to regularize the $w_k$ ranging in [0, 1]. The growth rate of the denominator is faster when the class size $n_k$ is larger, which leads to the motivation (1). In addition, when the actual class size $n_k$ is much smaller than expected size $n_k^e$, both the numerator and denominator approximate to one, which leads to a larger $w_k$ (i.e., $w_k$ = 1). This is consistent with the motivation (2). Moreover, when the $n_k$ is much larger than $n_k^e$, the $w_k$ is smaller than one. The result reflects the motivation (3). These explanations will be included in the revised version.
>
>
>
> Q3. Line 170: nominator -> numerator ?
>
> A3. We thank to the reviewer for a careful review. We will modify it in the revision.

---

> > ### Comment · Reviewer_Q3g7 · 2022-08-09
> > **comments on author response**
> >
> > Q&A1.  Thanks, I misread Lines 186-188.  Your revision with 2 separate sentences helps.
> >
> > Q&A2.  Fig 1 in the supplementary material: $w_k$ is exponentially decreasing.  Would a simpler expression such as $e^{-n_k}$ suffice?  Also, what are the considerations for $w_k$ when $n_k$ is within the lower and upper bounds?

---

> > > ### Author Response · Authors · 2022-08-10
> > > **Response to Reviewer 4**
> > >
> > > The responses to the two questions raised are as follows.
> > >
> > > Q1. Fig 1 in the supplementary material: $w_k$ is exponentially decreasing. Would a simpler expression such as $e^{-n_k}$ suffice?
> > >
> > > A1. Thanks for the question. We have plotted $w_k$ and $e^{-n_k}$ in the linked document: https://www.dropbox.com/s/p67lnevw3t7qqmr/response_to_reviewer4_additional_Q1.pdf?dl=0. It can be observed from the figures that when $n_k$ is large (i.e., majority classes), $w_k \simeq e^{-n_k}$. However, $e^{-n_k}$ is much smaller than $w_k$ when $n_k$ is small  (i.e., minority classes). Therefore, the $w_k$ cannot be fully expressed by $e^{-n_k}$.
> > >
> > > Moreover, we conducted the experiments on $w_k$ under different $\beta$ settings. As shown in the tables in the linked document, it can be seen that the performance is better when $\beta$ is larger, i.e., the weights on minority classes are smaller. Nevertheless, when $\beta = 0.999$, the performacne has been saturated. It indicates that it may have reached a good balance between the weights on the majority and the minority classes. Hence, $e^{-n_k}$ with much smaller weights on the minority classes than $\beta = 0.999$ may be subject to the accuracy degradation such as $\beta = 0.9999$.
> > >
> > > Q2. What are the considerations for $w_k$ when $n_k$ is within the lower and upper bounds?
> > >
> > > A2. When $n_k$ is within the lower and upper bounds, then the $k$-th class belongs to the moderated sized class (Case II.) as presented in Table 2 in Appendix. Then $w_k$ is designed between the weights on the majority classes (Case III.) and the weights on the minority classes (Case I.) since the $n_k$ is consistent with the hypothesis of the homogeneity of class variances (illustrated in Lines 65-66 in Appendix) which is described in Theorem 3.2 and stated in Lines 194-195. In contrast, the minority or majority classes whose sizes are out of bounds are regularized to much larger or smaller weights (see Table 2 in the Appendix).

---

### Official Review · Reviewer_Xgjj · 2022-07-11

**Rating:** 7
**Confidence:** 4
**Soundness:** 3 good
**Presentation:** 3 good
**Contribution:** 3 good

**Summary:**

The authors investigate quantization of (deep neural network) model parameters in class-imbalance settings. This is achieved by scaling the variance of of each class separately, which is then mapped to a common distribution for quantization. Further, a loss function with class reweighing is used to satisfy the homogeneity of class variances. Experiments on four datasets (Synddigit-LT, CIFAR-10-LT, CIFAR-100-LT, ImageNet-ILSVRC) demonstrate state-of-the-art results for quantized models, specially, on highly imbalanced data.

**Questions:**

Though the assumption of variance differences between classes of different sizes is reasonable and can be demonstrated in practice in multiple ways, using t-SNE is perhaps not the best way considering that due to the minimum distance specification in t-SNE, a larger dataset will occupy a larger space regardless of variance. Moreover, being a local linear embedding technique, distributional variance are not necessarily well represented. Can the authors justify or use a different visualization technique?

In Section 3.1.1 the authors claim that differences in class distributions cause massive quantization error, however, it is not justified or demonstrated.

There seems to be a problem in Figure 2, by which N of N(\mu_k,sigma_k) is not shown.

The authors argue that data follows a normal distribution, which is demonstrably not true in general practical settings (without data transformation). Though it is possible that the authors are referring to model parameters and features, the beginning of Section 3.1.2 clearly states that data x is assumed to be Gaussian, though later it is stated that x are features. Please clarify by stating the notation earlier.

What happens in practice when beta -> 1 (0.999 < beta < 1)?

Did the authors consider ClimbQ with a simple inverse probability weighting (1/p_k, where p_k is the proportion of samples in class k)?

**Limitations:**

The authors address the impact and limitations of the proposed approach in Section 6.

**Strengths And Weaknesses:**

Strengths:
Though the variance scaling and quantization are relatively straightforward, the weighting approach based on hypothesis testing offers a different and interesting angle to class weighting schemes. The experiments are extensive, consider multiple datasets, imbalance ratios and importantly, also consider existing class weighting approaches, which not originally intended for quantization, can be used within the ClimbQ framework.

Weaknesses:
Though the paper is relatively well organized, it is difficult to follow, for instance, N(.,.) is used before being defined, x is introduced as data to then being extended to also the features from a deep learning model, there is abuse of notation (e.g., F(X) a distribution is equated to a random variable U), x' is not formally defined, etc.

Something that is never explained or discussed is why if the data (including features) is being scaled during training, how is the scaling being applied during testing and if not necessary, why is it the case?

Post-rebuttal: increased rating after detailed clarification from the authors.

---

> ### Author Response · Authors · 2022-08-02
> **Response to Reviewer 3**
>
> We thank the reviewer for the careful reading of the manuscript and the constructive
> comments. The responses are as follows.
>
> W1. a) $N(.,.)$ is used before being defined, b) $x$ is introduced as data to then being extended to also the features from a deep learning model, c) there is abuse of notation (e.g., $F(X)$ a distribution is equated to a random variable $U$), d) $x'$ is not formally defined
>
> A1.
>
> a) Thank the reviewer for pointing out this. We will add the definition of the notation.
>
> b)  Thank you for your comment. We would like to modify the notation as follows. The network features for N data are denoted as $X =$ {$x_1, x_2, …, x_N$}.
> In probability, $F(X)$ generally represents a random variable that is transformed from the random variable $X$ through the function $F$, while $F(x)$ actually denotes the distribution value which can be derived from the probability, $P(X <= x)$.
> Thank you for pointing out this. To clarify the applied value of $x'$ in Eq. (2), we will add $x_k’ = D_k(x_k)$ below Eq. (2).
>
> W2. a) Why if the data (including features) is being scaled during training? b) How is the scaling being applied during testing and if not necessary, why is it the case?
>
> A2.
>
> a) The motivation of scaling has been illustrated in Lines 113-116. According to the exploration results in Fig. 1, we found that the quantization error is related to the class size and feature variation. Therefore, we aimed at scaling the class variations to the same scale for quantization.
>
> b) We also performed the scaling in testing. We took $\mu_k$ and $\sigma_k$ in Eq. (1) as the mean and standard deviation of the
> $k$-th class of testing data.
>
> Q1. Can the authors justify or use a different visualization technique?
>
> A1. Thanks for the suggestion. We calculate the actual class variances from the CNN features and visualize the results based on the bar chart. Please see the anonymous link: https://www.dropbox.com/s/b3ahyu07wjow0fb/response_to_reviewer3_Q1.pdf?dl=0.
>
> Q2. In Section 3.1.1, the authors claim that differences in class distributions cause massive quantization error, however, it is not justified or demonstrated.
>
> A2. In Lines 253-258 and Lines 270-273 of the main paper, it can be seen that the proposed ClimbQ is demonstrated with the higher accuracies, i.e., the smaller quantization errors, than the compared research in Table 1 and Table 2, since the difference of variations between class distributions is reduced.
>
> Q3. The $N$ of $N(\mu_k$,$\sigma_k)$ is not shown in Figure 2.
>
> A3. Thanks to the reviewer for pointing this out. This will be fixed in the revision.
>
> Q4. a) The normal distribution is demonstrably not true in general practical settings (without data transformation). b) The beginning of Section 3.1.2 clearly states that data x is assumed to be Gaussian, though later it is stated that x are features. Please clarify by stating the notation earlier.
>
> A4.
>
> a) We adopt the normal distributions in this paper mainly according to the exploration analyses on the benchmark datasets as shown in Fig. 2 in Appendix. If there is a dataset with another distribution as the reviewer mentioned, we can replace the distribution function (DF) of normal distribution in Eq. (1) with the DF of the new distribution since the $X$ in Theorem 1 can be a random variable of any continuous distribution.
>
> b) Thank you for your comment. We shall modify the notation as follows and will state earlier in the revision. The network features for N data are denoted as $X$ = {$x_1$, $x_2$, …, $x_N$}.
>
> Q5. What happens in practice when $\beta$ -> 1 (0.999 < $\beta$ < 1)?
>
> A5. Thanks for the question. We have conducted the experiments by setting $\beta$ = 0.9999. The results are presented through the following anonymous link: https://www.dropbox.com/s/vw7ukd9bf1wzr3e/response_to_reviewer3_Q5.pdf?dl=0. From the linked document, although we can obtain a better performance when we set a larger $\beta$ value, the accuracy gets saturated when $\beta$ approximates to one, e.g., $\beta$ >= 0.999, since the weight on the majority approximates to zero (see Fig. 1 in Appendix).
>
> Q6. Did the authors consider ClimbQ with a simple inverse probability weighting ($1/p_k$, where $p_k$ is the proportion of samples in class $k$)?
>
> A6. We have considered $1/p_k$ ($1/p_k = n/n_k$), where $n = (n_1 + n_2 + … + n_k)$. However, such a design did not work well since $1/p_k$ can be extremely large or small if the data are highly imbalanced, e.g., suppose $n = n_1 + n_2 = 9999 + 1$, then $w_1 = 10000/9999 ~= 1$, and $w_2 = 10000/1 = 10000$. Accordingly, the training process is unstable.
>
> Therefore, in this paper, we designed $w_k$ with $\beta = 0.999$ as the base of an exponent (see the design of w_k in Line 212) which restricts $w_k$ not in a large range. The $w_k$ is analyzed in Appendix A.3, and the result in Fig. 1 in Appendix shows that the $w_k$ is limited in the range [0, 1]. The effectiveness of the design is validated in the experiments (see Sec. 4.2).

---

> > ### Author Response · Authors · 2022-08-07
> > **Response to Reviewer 3**
> >
> > We number c) and d) in the response to W1 as below.
> >
> > W1. a) $N(.,.)$ is used before being defined, b) $x$ is introduced as data to then being extended to also the features from a deep learning model, c) there is abuse of notation (e.g., $F(X)$ a distribution is equated to a random variable $U$), d) $x'$ is not formally defined
> >
> > A1.
> >
> > a) Thank the reviewer for pointing out this. We will add the definition of the notation.
> >
> > b)  Thank you for your comment. We would like to modify the notation as follows. The network features for N data are denoted as $X =$ {$x_1, x_2, …, x_N$}.
> >
> > c) In probability, $F(X)$ generally represents a random variable that is transformed from the random variable $X$ through the function $F$, while $F(x)$ actually denotes the distribution value which can be derived from the probability, $P(X <= x)$.
> >
> > d) Thank you for pointing out this. To clarify the applied value of $x'$ in Eq. (2), we will add $x_k’ = D_k(x_k)$ below Eq. (2).

---

> > ### Comment · Reviewer_Xgjj · 2022-08-08
> > **Additional questions**
> >
> > Thanks for the detailed answers. The reviewer still has two questions:
> >
> > 1) Following up on the scaling of the test data. The question is how to perform: "We took mu_k and sigma_k in Eq. (1) as the mean and standard deviation of the k-th class of testing data", if during testing the class labels are not available?
> >
> > 2) About the massive quantization error not being demonstrated or justified. The reviewer means the actual quantization error. Based on the paper, the authors seem to imply that the accuracy differences are caused by quantization errors but the extent of the quantization error is not demonstrated, so the claim sounds more like a hypothesis than a fact.

---

> > > ### Author Response · Authors · 2022-08-09
> > > **Response to Reviewer 3**
> > >
> > > The responses to the two questions raised are as follows.
> > >
> > > Q1. Following up on the scaling of the test data. The question is how to perform: "We took mu_k and sigma_k in Eq. (1) as the mean and standard deviation of the k-th class of testing data", if during testing the class labels are not available?
> > >
> > > A1. Thanks for the question. The implementation details are as follows.
> > >
> > > Basically, the testing data can be divided into the validation set (for the decision of hyperparameters) and the testing set (for inference and performance evaluation).
> > >
> > > For the testing data in the validation set (where the class labels are available), we took $\mu_k$ and $\sigma_k$ as the mean and standard deviation of the $k$-th class. In addition, the scale factor $c_k$ defined in Line 119 can also be obtained since the sizes of classes are known.
> > >
> > > On the other hand, since the class information was unknown for the other testing data for inference, we adopted the distribution function of the standard normal, $N(0, 1)$, instead of $N(\mu_k, c_k^2 \sigma_k^2)$, in our experiments. Note that the main reason that we utilized the standard normal is that the features had been normalized through the batch normalization layer before quantization. These explanations will be added to our revised version for better readability.
> > >
> > > Moreover, the learning and inference processes have been empirically validated in our experiments (see Sec. 4), indicating that the quantized model used can learn the identification capability well on the imbalance data during the inference.
> > >
> > >
> > >
> > > Q2. About the massive quantization error not being demonstrated or justified. The reviewer means the actual quantization error. Based on the paper, the authors seem to imply that the accuracy differences are caused by quantization errors but the extent of the quantization error is not demonstrated, so the claim sounds more like a hypothesis than a fact.
> > >
> > > A2. We have examined the actual quantization errors, and the results are presented through the anonymous link: https://www.dropbox.com/s/n4295bndnko193t/response_to_reviewer3_additional_Q2.pdf?dl=0.
> > >
> > > The quantization error is measured as: $\frac{|w_q - w|}{max(w_q) - min(w_q)}$. The numerator is the discrepancy between the quantized weight $w_q$ and the original floating-point weight $w$. We further normalized the error by dividing the range of the quantized space.
> > >
> > > As shown in the linked document, we compared the actual quantization errors with and without the proposed scaling approach (see Sec. 3.1). It can be observed from the results that when the class distributions are not scaled (denoted as “w/o scaling”), i.e., the differences in the class distributions are not reduced, we have a larger quantization error and therefore a lower accuracy. In contrast, after the class distributions are scaled to the similar variations (denoted as “w/. scaling”) and projected to the same space before the quantization (see Sec. 3.1), the total quantization error is reduced, hence leading to a higher accuracy. The relationship between the quantization error and the accuracy is then illustrated in Lines 114-115. The charts presented in the linked document will be added to the Appendix in the revised version.

---

> > > > ### Comment · Reviewer_Xgjj · 2022-08-10
> > > > **Response**
> > > >
> > > > Thanks for the detailed answer and the new quantization error results. I have updated the rating accordingly.

---

### Official Review · Reviewer_h4jR · 2022-07-11

**Rating:** 5
**Confidence:** 3
**Soundness:** 3 good
**Presentation:** 3 good
**Contribution:** 2 fair

**Summary:**

This paper investigates the issue of quantization on class-imbalanced data. The key observation in this work is that quantizing imbalanced data inclines to obtain a large error due to the differences between separate class distributions, which leads to a significant accuracy loss. In order to tackle this issue, this work proposes ClimbQ, a new framework that focuses on diminishing the inter-class heterogeneity for quantization error reduction. ClimbQ first scales the variance of each class distribution and then projects data through the new distributions to the same space for quantization. To guarantee the homogeneity of class variances after the ClimbQ process, we examine the quantized features and derive that the homogeneity satisfies when data size for each class is restricted (bounded). Accordingly, ClimbQ embeds a new Homogeneous Variance Loss (HomoVar Loss), which reweights the data losses of each class based on the bounded data sizes to satisfy the homogeneity of class variances. Extensive experiments on class-imbalanced and benchmark balanced datasets reveal that ClimbQ outperforms the state-of-the-art quantization techniques, especially on highly imbalanced data

**Questions:**

- What are the benefits (and/or downsides) of ClimbQ over other mixed-quantization schemes?
- Can you comment on potential applicability of your approach on post-training quantization?
- Can you comment on potential impacts of your approach on the inference performance?

**Limitations:**

- I assume the work requires the knowledge of the distribution on the full dataset (which may limit the practicality of this approach since it may require multi-party sharing).
- I am concerned about the potential implications in terms of the inference performance.
- I would suggest to make the scope of your approach clear upfront.

**Strengths And Weaknesses:**

Strengths:
- Important problems and the insights are valuable
- An end-to-end solution to address the class-imbalanced issues within quantization
- Evaluation results are sufficient to justify the effectiveness of the proposed approach
- Source codes available

Weaknesses:
- Limited novelty (the solution donot add new insights on class-imbalanced quantization)
- The proposed solution seems restricted within quantization-aware training

---

> ### Author Response · Authors · 2022-08-02
> **Response to Reviewer 2**
>
> We thank the reviewer for taking the time for a careful review. The responses are listed as follows.
>
>
> W1. Limited novelty (the solution do not add new insights on class-imbalanced quantization)
>
> A1. In this paper, we adopt the “distribution function” (defined in Eq. (1)) to generate scaled class feature distributions for quantization to reduce the quantization errors of minority classes (illustrated in Sec. 3.1). Prior quantization research (see Sec. 2) has not considered the impact of the heterogeneity of class distributions on the quantization results and performance. In addition, to the best of our knowledge, the distribution function has not been utilized to scale the distributions of network features. Hence, in our opinion, our approach possesses novelty and new insights from these perspectives.
>
>
> W2. The proposed solution seems restricted within quantization-aware training
>
> A2. We mainly focus on the Quantization-Aware Training (QAT) research to design the approaches since QAT with the fine-tuning process usually outperforms another branch Post-training Quantization (PTQ) [1]. In addition, QAT learns the quantized models more efficiently without the information of full-precision models that Zero-shot Quantization (ZSQ) research requires [1].
>
>
>
> Q1. What are the benefits (and/or downsides) of ClimbQ over other mixed-quantization schemes?
>
> A1. ClimbQ is designed with the consideration of the heterogeneity of the class distributions to address the imbalance issue. In contrast, existing mixed-precision quantization schemes without exploring the imbalance problem tend to suffer from an issue of a large quantization error in the minority classes (as presented in Fig. 1). Nevertheless, mixed-precision quantization schemes can flexibly assign the required bits to each network layer. Therefore, assuming data are balanced, mixed-precision quantization schemes are generally able to obtain a better prediction performance than fixed-precision schemes [1], such as ClimbQ.
>
>
> Q2. Can you comment on potential applicability of your approach on post-training quantization?
>
> A2. The proposed approach can conceptually be applicable on the imbalance post-training quantization. We can first remove the component of uniform quantization and pretrain a full-precision model only using the generation of scaled class distribution (see Sec. 3.1) and the HomoVar rebalancing loss (see Sec. 3.2). During the quantization process, we retain the scaled class distribution component (since the learned weights are based on the new distributions) and then implement the uniform quantization (see Eq. (2)) or other existing PTQ quantization functions.
>
>
> Q3. Can you comment on potential impacts of your approach on the inference performance?
>
> A3. Previous QAT and PTQ research used trained quantization parameters in inference. However, there is generally a difference between the training and testing distributions in the real data, which may lead to a bias during inference and the accuracy degradation. In contrast, we utilize the testing statistics (e.g., mean and variance) in inference, thereby addressing the issue of the inference bias.
>
>
> L1. I assume the work requires the knowledge of the distribution on the full dataset (which may limit the practicality of this approach since it may require multi-party sharing).
>
> A1. Thanks for the point raised. However, there may be a misunderstanding which needs to be clarified. In this paper, we adopted a batch-wise processing in inference. Sec. 4 shows the effectiveness of ClimbQ in the case that only a few testing data (batch data) are accessed. As such, we can independently process the multi-party data during inference.
>
> L2. I am concerned about the potential implications in terms of the inference performance.
>
> A2. This is same as the question 3. Please see the response to Q3. Thanks.
>
> L3. I would suggest to make the scope of your approach clear upfront.
>
> A3. Thanks for the suggestion. We will clarify the scope upfront in the revision.
>
>
> Reference
> - [1] Gholami, A., Kim, S., Dong, Z., Yao, Z., Mahoney, M. W., & Keutzer, K. (2021). A survey of quantization methods for efficient neural network inference. arXiv preprint arXiv:2103.13630.

---

> > ### Comment · Reviewer_h4jR · 2022-08-10
> > **Thanks for your detailed response**
> >
> > Thanks for your detailed responses, and most of my concerns are already resolved. (including W2, Q2, L1, Q3, L2). I would consider raising the score if the rest of comments are addressed with the revision ready.
> >
> > (1) I am wondering the opinions from the authors on the potential synergy between ClimbQ and mixed-precision quantization. From my perspective, ClimbQ can be potentially combined with mixed-precision quantization since the imbalanced class distribution can also make the range of the value representation divergent. Do you think it's feasible to potentially combine these two lines of works?
> >
> > (2) For inference performance, I am referring to the inference latency/throughput. Can ClimbQ help with the inference performance by exploiting class imbalance?

---

> > > ### Author Response · Authors · 2022-08-10
> > > **Response to Reviewer 2**
> > >
> > > The responses to the two questions raised are as follows.
> > >
> > > Q1. I am wondering the opinions from the authors on the potential synergy between ClimbQ and mixed-precision quantization. From my perspective, ClimbQ can be potentially combined with mixed-precision quantization since the imbalanced class distribution can also make the range of the value representation divergent. Do you think it's feasible to potentially combine these two lines of works?
> > >
> > > A1. Yes, we also consider that the mixed-precision quantization may be applicable to the imbalanced class distributions with different ranges. The classes with larger ranges can be assigned with more bits (i.e., using more quantized values), and the classes with smaller ranges can be assigned with fewer bits (i.e., using fewer quantized values) to effectively reduce the quantization errors $|x - Q(x)|$ and avoid a significant performance degradation according to [1]. In addition to the range, we also think that it may also be feasible to utilize other metrics such as the Hessian matrix and eignenvalues [2, 3] to measure the contained information in separate class distributions for the decision of the assignment of bits.
> > >
> > > --Reference
> > > - [1] Rastegari, M., Ordonez, V., Redmon, J., & Farhadi, A. (2016, October). Xnor-net: Imagenet classification using binary convolutional neural networks. In European conference on computer vision (pp. 525-542). Springer, Cham.
> > > - [2] Shen, S., Dong, Z., Ye, J., Ma, L., Yao, Z., Gholami, A., ... & Keutzer, K. (2020, April). Q-bert: Hessian based ultra low precision quantization of bert. In Proceedings of the AAAI Conference on Artificial Intelligence (Vol. 34, No. 05, pp. 8815-8821).
> > > - [3] Wang, K., Liu, Z., Lin, Y., Lin, J., & Han, S. (2019). Haq: Hardware-aware automated quantization with mixed precision. In Proceedings of the IEEE/CVF Conference on Computer Vision and Pattern Recognition (pp. 8612-8620).
> > >
> > > Q2. For inference performance, I am referring to the inference latency/throughput. Can ClimbQ help with the inference performance by exploiting class imbalance?
> > >
> > > A2. Thanks for the question. We have conducted experiments to compare the inference time (sec./batch) and throughput (#images/sec.) of ClimbQ with the quantization baselines in the paper. The results are presented in the linked document: https://www.dropbox.com/s/nqxps1p3sjbeo9c/response_to_reviewer2_additional_Q2.pdf?dl=0.
> > >
> > > It can be observed that ClimbQ has fewer time costs in inference, i.e., with smaller latency compared with other approaches. In addition, the throughput of ClimbQ is higher, i.e., more images can be processed in a fixed time span. The better efficiency of ClimbQ in inference than that of the compared approaches is mainly due to a simple function adopted (see Eq. (1)) for the scaling and projection of class distributions and the uniform quantization (see Eq. (2)) without other additional operations used in the compared approaches, such as clipping functions and transformations.

---

> > > > ### Comment · Reviewer_h4jR · 2022-08-10
> > > > **Score Raised based on the rebuttal**
> > > >
> > > > Thanks for your detailed elaboration. I recommend the authors to combine the above content into the paper, since it can strengthen the contributions of your work. I do appreciate the efforts and therefore I raised the score to 5.

---

### Official Review · Reviewer_bnA4 · 2022-07-17

**Rating:** 7
**Confidence:** 3
**Soundness:** 3 good
**Presentation:** 4 excellent
**Contribution:** 3 good

**Summary:**

The authors propose better quanitization techniques for class-imbalanced datasets.

Specifically, propose
1. ClimbQ - scale the class distributions such that rare classes have higher variance, followed by bucketization of CDF
2. ClimbQ + homovar - a weighted loss function inspired from Levene’s hypothesis testing.

The authors compare the proposal against related techniques on 3 datasets, mostly by simulating different levels of imbalance.


**Questions:**

- Have the authors considered plugging in eq(4) to other competing approaches?

**Strengths And Weaknesses:**

Strengths.
---------

- The proposal is well-motivated, simple and shows strong experimental results.
- The paper is also well-written.

Weakness
--------
- The generation of simulated data through γ is a bit unconvincing since it was not clear how exactly was the imbalance simulated.
  Having said that, I do think results in Appendix#3 sound convincing.

---

> ### Author Response · Authors · 2022-08-02
> **Response to Reviewer 1**
>
> We thank the reviewer for taking the time to review our paper. In response to the comments, our replies are as follows.
>
> W1. The generation of imbalance data using γ  is unclear.
>
> A1. In this paper, we follow the previous research on imbalance (long-tailed) learning [1, 2, 3] to generate the imbalance data by the ratio γ. As described in Lines 230-231,  the imbalance ratio γ indicates the number of the largest training class divided by that of the smallest. Below is the detailed procedure, by taking CIFAR-10-LT with γ = 50 for an example.
>
> - Step 1. We sampled from the balanced data CIFAR-10 in which training data instances were 5000 for each class and a total of 10 classes are indexed (labeled) from 0 to 9.
> - Step 2. We chose the class 0 as the maximal class with size 5000.
> - Step 3. We chose the class 9 as the minimal class and sampled 5000/γ = 5000/50 = 100 data.
> - Step 4. The other classes (indexed 1 to 8) were then sampled in exponential distribution as shown in Fig 1. (a).
>
>
> Q1. Have the authors considered plugging in eq(4) to other competing approaches?
>
> A1. Thanks to the reviewer for the suggestion to implement the HomoVar loss on other competing approaches. We have conducted the experiments and the results are provided through the anonymous link: https://www.dropbox.com/s/9afaq6fymqv0hbg/response_to_reviewer1_Q1.pdf?dl=0. It can be seen that the HomoVar loss incorporated with our proposed ClimbQ quantization process can still achieve the best performances since the compared works did not fully explore the difference between the class distributions.
>
> Reference
> - [1] Ren, Jiawei, Cunjun Yu, Xiao Ma, Haiyu Zhao, and Shuai Yi. "Balanced meta-softmax for long-tailed visual recognition." Advances in neural information processing systems 33 (2020): 4175-4186.
> - [2] Li, M., Cheung, Y. M., & Lu, Y. (2022). Long-tailed Visual Recognition via Gaussian Clouded Logit Adjustment. In Proceedings of the IEEE/CVF Conference on Computer Vision and Pattern Recognition (pp. 6929-6938).
> - [3] Wei, C., Sohn, K., Mellina, C., Yuille, A., & Yang, F. (2021). Crest: A class-rebalancing self-training framework for imbalanced semi-supervised learning. In Proceedings of the IEEE/CVF conference on computer vision and pattern recognition (pp. 10857-10866).

---

### Meta-Review · Area_Chair_JdSi · 2022-08-23

**Recommendation:** Accept
**Confidence:** Certain

**Metareview:**

After rebuttal, the reviewers unanimously agree that the submission should be accepted for publication at NeurIPS.

**Award:**

No

---

### Decision · Program_Chairs · 2022-09-14

Accept